# Control of feeding by Piezo-mediated gut mechanosensation in *Drosophila*

**Soohong Min[1], Yangkyun Oh[2], Pushpa Verma[3], Samuel C Whitehead[4], Nilay Yapici[5], David Van Vactor[3], Greg SB Suh[2,6], Stephen Liberles[1]\***

[1]Howard Hughes Medical Institute, Harvard Medical School, Department of Cell Biology, Boston, United States; [2]Skirball Institute, NYU School of Medicine, New York, United States; [3]Harvard Medical School, Department of Cell Biology, Boston, United States; [4]Department of Physics, Cornell University, Ithaca, United States; [5]Department of Neurobiology and Behavior, Cornell University, Ithaca, United States; [6]KAIST, Department of Biological Sciences, Daejeon, Republic of Korea

**Abstract** Across animal species, meals are terminated after ingestion of large food volumes, yet underlying mechanosensory receptors have so far remained elusive. Here, we identify an essential role for *Drosophila* Piezo in volume-based control of meal size. We discover a rare population of fly neurons that express Piezo, innervate the anterior gut and crop (a food reservoir organ), and respond to tissue distension in a Piezo-dependent manner. Activating Piezo neurons decreases appetite, while *Piezo* knockout and Piezo neuron silencing cause gut bloating and increase both food consumption and body weight. These studies reveal that disrupting gut distension receptors changes feeding patterns and identify a key role for *Drosophila* Piezo in internal organ mechanosensation.

## Introduction

Mechanosensory neurons detect a variety of environmental forces that we can touch or hear, as well as internal forces from organs and tissues that control physiological homeostasis (*Abraira and Ginty, 2013*; *Ranade et al., 2015*; *Umans and Liberles, 2018*). In many species, specialized mechanosensory neurons innervate the gastrointestinal tract and are activated by tissue distension associated with consuming a large meal (*Williams et al., 2016*; *Zagorodnyuk et al., 2001*). Gut mechanosensation may provide an evolutionarily conserved signal for meal termination as gut distension inhibits feeding in many species and evokes the sensation of fullness in humans (*Phillips and Powley, 1996*; *Rolls et al., 1998*). However, how gut distension receptors contribute to long-term control of digestive physiology and behavior is unclear as tools for selective pathway manipulation are lacking. Identifying neuronal mechanisms involved in detecting the volume of ingested food would provide basic insights into this fundamental mechanosensory process, and in humans, perhaps clinical targets for feeding and metabolic disorders.

Here, we investigated the roles and mechanisms of food volume sensation in the fruit fly *Drosophila melanogaster*. Volumetric control of feeding was classically studied in a larger related insect, the blowfly, with relevant mechanosensory hotspots identified in the foregut and crop, an analog of the stomach (*Dethier and Gelperin, 1967*; *Gelperin, 1967*). In *Drosophila*, chemosensory neurons detect nutrients in the periphery and brain to control appetite, with some neurons positively reinforcing feeding during starvation conditions (*Bjordal et al., 2014*; *Dus et al., 2015*; *Miyamoto et al., 2012*). In contrast, the importance of gut mechanosensation in *Drosophila* feeding control and digestive physiology has not been similarly investigated; mechanosensory neurons of the gustatory system sense food texture and modulate ingestion (*Sánchez-Alcañiz et al., 2017*; *Zhang et al., 2016*), and other mechanosensory neurons in the posterior gut control defecation and

**\*For correspondence:**
Stephen_Liberles@hms.harvard.edu

food intake (*Olds and Xu, 2014*; *Zhang et al., 2014*). In contrast, food storage during a meal occurs primarily in the anterior gut (*Lemaitre and Miguel-Aliaga, 2013*; *Stoffolano and Haselton, 2013*). Enteric neurons of the hypocerebral ganglion innervate the fly crop, foregut, and anterior midgut, and lesioning of the recurrent nerve (which contains neurons of the hypocerebral ganglion) in *Drosophila* and blowfly increases feeding duration (*Dethier and Gelperin, 1967*; *Gelperin, 1967*; *Pool et al., 2014*). Together, these prior studies raise the possibility that a subpopulation of enteric neurons in *Drosophila* could be specialized to sense meal-associated gut distension.

## Results and discussion

### Piezo-expressing enteric neurons innervate the gastrointestinal tract

To explore whether food volume sensation occurs in *Drosophila* and to investigate underlying mechanisms, we first asked whether neurons expressing various mechanosensory ion channels innervated the anterior gut. Several mechanosensitive ion channels have been reported in *Drosophila*, including TRP channels (Nompc, Nanchung, and Inactive), the degenerin/epithelial sodium channel Pickpocket (Ppk), transmembrane channel-like (Tmc) protein, and Piezo (*Coste et al., 2012*; *Montell, 2005*; *Zhang et al., 2016*; *Zhong et al., 2010*). We obtained Gal4 driver lines that mark neurons containing mechanoreceptor proteins or related family members, induced expression of membrane-tethered CD8-Green Fluorescent Protein (GFP) or dendritically targeted DenMark fluorescent reporters, and visualized neuronal innervation of the anterior gut. We observed a small group of Piezo-expressing enteric neurons located in the hypocerebral ganglion (~5–6 neurons per fly), and a dense network of Piezo fibers throughout the crop and anterior midgut (*Figure 1A and B*). Hypocerebral ganglion neurons were similarly labeled and anterior gut innervation similarly observed in three independent *Piezo-Gal4* driver lines (*Figure 1—figure supplement 1A*), but not in other Gal4 lines analyzed. We noted Nanchung expression in some epithelial cells of the crop duct, but not in crop-innervating neurons. The hypocerebral ganglion and adjacent corpora cardiaca together contain ~35 neurons per fly based on Elav immunohistochemistry, and Piezo neurons therein were distinct from other neurons that expressed the fructose receptor Gr43a (~5 neurons per fly) or the glucagon analog adipokinetic hormone (Akh, ~20 neurons per fly) (*Figure 1C, D*, *Figure 1—figure supplement 1B*). Piezo neurites formed a muscle-associated lattice in the gut, and ascending axons contributed to the recurrent nerve (*Figure 1E, F*). Using a genetic approach involving the MultiColor FlpOut system (*Nern et al., 2015*) for sparse labeling of Piezo cells, flies were obtained with reporter expression in one or a few hypocerebral ganglion neurons but not in brain structures such as the pars intercerebralis; in these flies, separate Piezo neurons were observed to innervate the crop and/or anterior midgut (*Figure 1—figure supplement 1C*). *Drosophila* Piezo was previously shown to confer mechanically activated currents when expressed in human cells and to mediate mechanical nociception (*Coste et al., 2012*; *Kim et al., 2012*). Furthermore, vertebrate Piezo homologs play diverse mechanosensory roles, including in internal sensation of airway volume and blood pressure (*Min et al., 2019*; *Nonomura et al., 2017*; *Zeng et al., 2018*). We hypothesized that *Drosophila* enteric neurons that express Piezo and innervate the anterior gut might mediate volumetric control of appetite.

### Piezo neurons control feeding behavior

To explore this model, we activated and silenced Piezo neurons using genetic approaches and monitored feeding behavior. We expressed temperature-sensitive Shibire (Shi^ts) that blocks synaptic transmission at non-permissive temperatures (>32℃) in Piezo neurons using three independent *Piezo-Gal4* drivers (*Piezo>Shi^ts*). *Piezo>Shi^ts* flies were reared at a permissive temperature (18℃) and later tested for physiological and behavioral changes at 32℃. To measure feeding behavior, flies were fasted for 24 hr, and then given brief access (30 min) to food containing a dye for visualization and quantification of ingestion (*Figure 2A*). *Piezo>Shi^ts* flies from all three genotypes fed ravenously, and histological examination of the gastrointestinal tract showed gut bloating with increased crop size (*Figure 2B*). For comparison, genetic silencing of other gut-innervating neurons labeled in *GMR51F12-Gal4* flies (*Figure 2—figure supplement 1*) did not impact appetite or cause crop distension. These findings indicate that disrupting Piezo neurons compromises gut volume homeostasis and associated control of feeding.

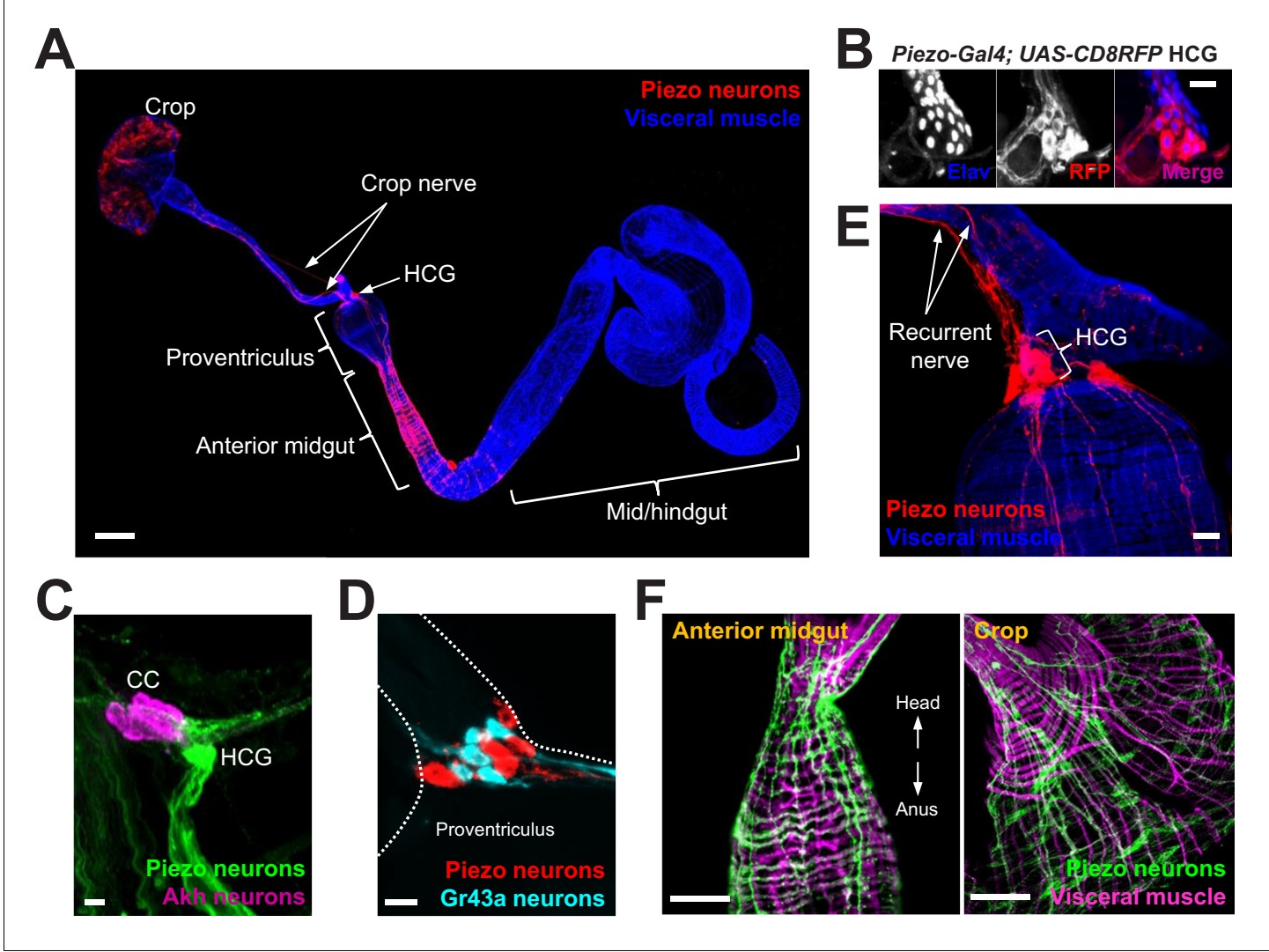

**Figure 1.** Piezo neurons innervate the gastrointestinal tract. (**A**) Wholemount image of the digestive tract from a *Piezo-Gal4 (59266); UAS-DenMark* fly visualized with immunofluorescence for DenMark (red, anti-Red Fluorescent Protein or RFP) and a fluorescent Phalloidin conjugate (blue) to label visceral muscle. HCG: hypocerebral ganglion, scale bar 100 µm. (**B**) Immunofluorescence for RFP (red) and Elav (blue) in the HCG from a *Piezo-Gal4; UAS-CD8RFP* fly, scale bar 10 µm. (**C**) Immunofluorescence for GFP (green) and Akh (magenta) in the corpora cardiaca (CC) and HCG from a *Piezo-Gal4; UAS-CD8GFP* fly, scale bar 10 µm. (**D**) Native GFP and RFP fluorescence from the HCG of a *Piezo-Gal4; UAS-CD8RFP; Gr43a-LexA; LexAop-CD8GFP* fly, scale bar 10 µm. (**E**) Image of the recurrent nerve (arrows) labeled by native RFP fluorescence in a *Piezo-Gal4; UAS-CD8RFP* fly and a fluorescent Phalloidin conjugate (blue), scale bar 10 µm. (**F**) The anterior midgut (left) and crop (right) of a *Piezo-Gal4; UAS-DenMark* fly visualized by immunofluorescence for DenMark (green) and a fluorescent Phalloidin conjugate (magenta), scale bar 50 µm. See *Figure 1—figure supplement 1* and source data.

The online version of this article includes the following source data and figure supplement(s) for figure 1:

**Figure supplement 1.** Innervation of the gastrointestinal tract by Piezo neurons.

**Figure supplement 1—source data 1.** Numerical data to support the graph in *Figure 1—figure supplement 1*.

To test the effects of activating Piezo neurons on food consumption, we drove expression of the temperature-regulated ion channel Trpa1 in Piezo neurons using *Piezo-Gal4* lines (*Piezo>Trpa1*). Thermogenetic activation of Trpa1 in Piezo cells, achieved by transferring *Piezo>Trpa1* flies from 18°C to 30°C, suppressed food intake after a 24-hr fast and also blocked meal-associated increases in crop volume, with similar results observed using three different *Piezo-Gal4* drivers (*Figure 2C*). Since many cell types express Piezo (*Kim et al., 2012*), we next used approaches for intersectional genetics involving Gal80, a dominant suppressor of Gal4-mediated gene induction to restrict Trpa1

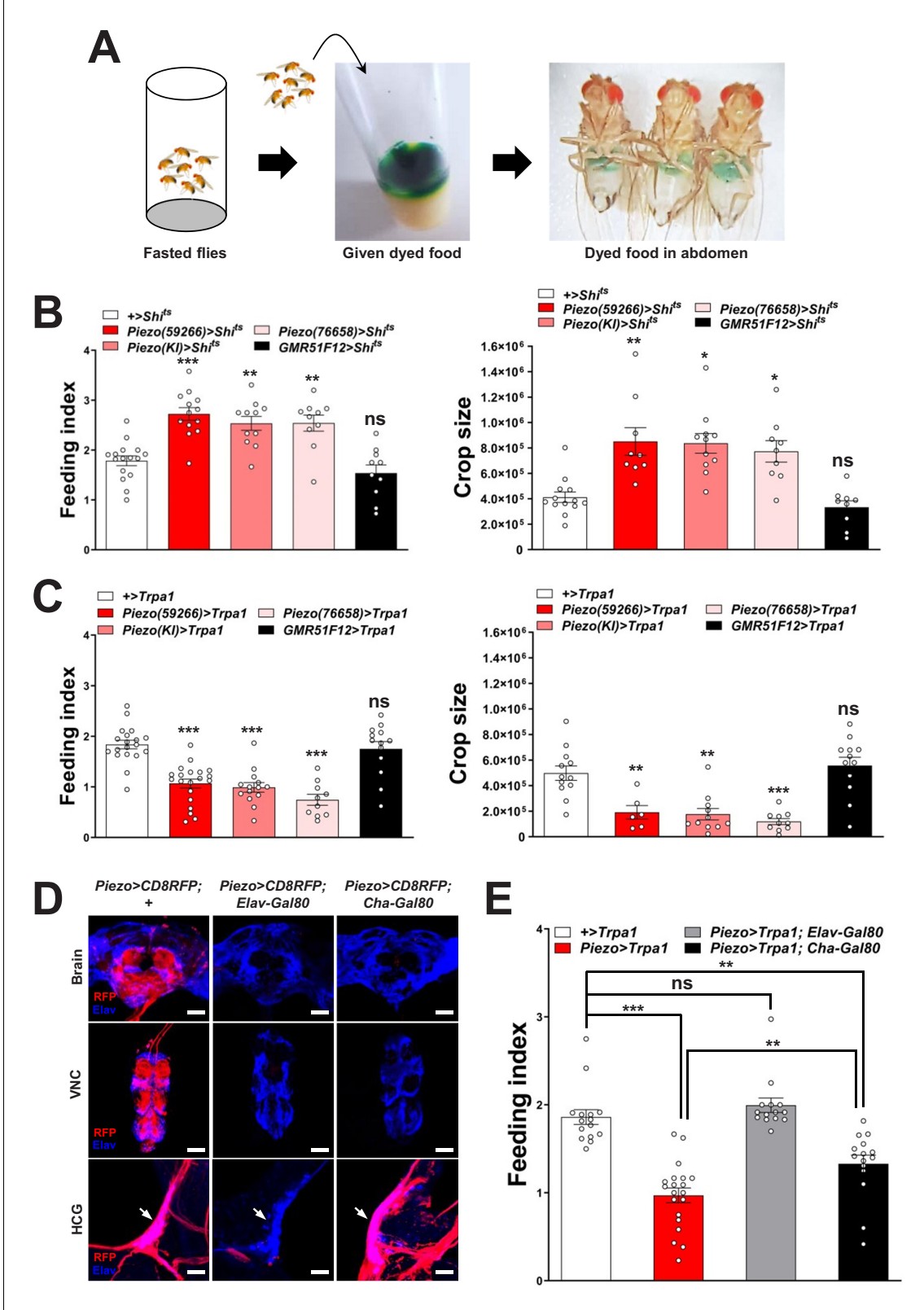

**Figure 2.** Piezo neurons control feeding behavior. (**A**) Depiction of the colorimetric feeding assay. (**B**) Fasted flies with *Shibire* alleles indicated were given brief access (30 min) to dye-labeled food at 32°C, and feeding indices and crop sizes were calculated. n (left to right) (feeding index): 16, 11, 13, 10, and 10 trials involving 12 flies per trial. n (crop size): 13, 9, 11, 9, and 9 flies, mean ± SEM, ***p<0.0005, **p<0.005, *p<0.05, ns: not significant by ANOVA Dunnett's multiple comparison test. (**C**) Fasted flies with *Trpa1* alleles indicated were given brief access (30 min) to dye-labeled food at 30°C,

*Figure 2 continued on next page*

*Figure 2 continued*

and feeding indices and crop sizes were calculated. n (left to right) (feeding index): 19, 20, 14, 10, and 13 trials involving 12 flies per trial. n (crop size): 12, 6, 11, 10, and 12 flies, mean ± SEM, \*\*\*p<0.0005, \*\*p<0.005, ns: not significant by ANOVA Dunnett's multiple comparison test. (D) Native RFP fluorescence in brain (top), ventral nerve cord (VNC, middle), and hypocerebral ganglion (HCG, bottom) of *Piezo-Gal4*[59266]; *UAS-CD8RFP* flies with *Gal80* alleles indicated, scale bar 100 μm (brain, VNC), 20 μm (HCG). (E) Fasted flies with *Trpa1* alleles indicated were given brief access (30 min) to dye-labeled food at 30°C, and feeding indices were calculated. n (left to right): 15, 20, 14, and 15 trials involving 12 flies per trial, mean ± SEM, \*\*\*p<0.0005, \*\*p<0.005, ns: not significant by ANOVA Dunnett's multiple comparison test. See *Figure 2—figure supplements 1–3* and source data.

The online version of this article includes the following source data and figure supplement(s) for figure 2:

**Source data 1.** Numerical data to support the graphs in *Figure 2*.
**Figure supplement 1.** Visualizing gut innervation by neurons labeled in *GMR51F12-Gal4* flies.
**Figure supplement 2.** Visualizing and manipulating subtypes of Piezo neurons.
**Figure supplement 2—source data 1.** Numerical data to support the graph in *Figure 2—figure supplement 2*.
**Figure supplement 3.** Analyzing central and peripheral cell types labeled in various genetic models.
**Figure supplement 3—source data 1.** Numerical data to support the graph in *Figure 2—figure supplement 3*.

expression to fewer cells. First, we drove Gal80 expression broadly in neurons using *Piezo>Trpa1; Elav-Gal80* flies and observed restoration of normal feeding behavior, indicating the relevant Piezo expression site to be neurons (*Figure 2D, E*). Among neurons, *Piezo-Gal4* drove expression in various peripheral sensory neurons, the ventral nerve cord, brain, and hypocerebral neurons. Differential expression control could be partially achieved using a *Cha-Gal80* driver, which silences Gal4-mediated expression in the ventral nerve cord and many central neurons, but not in gut-innervating hypocerebral neurons or a few cells of the proboscis, intestine, and brain (*Figure 2D*, *Figure 2—figure supplement 2A*). Thermogenetic experiments in *Piezo>Trpa1; Cha-Gal80* flies also caused robust suppression of feeding behavior (*Figure 2E*). Intestinal cells are unlikely to contribute to feeding phenotypes in *Piezo>Trpa1; Cha-Gal80* flies based on experiments involving *Piezo>Trpa1; Elav-Gal80* flies; to provide additional evidence, we obtained *Escargot-Gal4* flies in which Piezo-expressing intestinal stem cells (ISCs) are broadly marked (*He et al., 2018*) and found that thermogenetic activation of intestinal cells using *Escargot-Gal4; UAS-Trpa1* flies also had no effect on feeding (*Figure 2—figure supplement 2B, C*). Piezo neurons expressing Dilp2 in the pars intercerebralis are also reported to innervate the crop and control feeding behavior (*Wang et al., 2020*), which potentially explain the significant differences we observe in feeding following thermogenetic activation experiments involving *Piezo-Gal4; UAS-Trpa1* and *Piezo-Gal4; UAS-Trpa1; Cha-Gal80* flies (*Figure 2E*). In control *Piezo-Gal4; UAS-CD8RFP* flies, we observed reporter expression per fly in 6.2 ± 0.5 hypocerebral neurons and 4.9 ± 1.0 pars intercerebralis neurons, 2.9 ± 0.7 of which express Dilp2. In *Piezo-Gal4; UAS-CD8RFP; Cha-Gal80* flies, we observed reporter expression per fly in 5.2 ± 0.5 hypocerebral neurons and 1.1 ± 0.5 pars intercerebralis neuron, 0.4 ± 0.3 of which express Dilp2 (about half of flies had one co-labeled neuron and half had zero) (*Figure 2—figure supplement 3A–C*). In flies that lacked any reporter expression in pars intercerebralis Dilp2 neurons, we still observed labeled neurites in the anterior midgut and crop nerve, consistent with findings from stochastic labeling (*Figure 1—figure supplement 1C*) that neurons outside of the pars intercerebralis innervate these regions. Furthermore, *Dilp2-Gal4* does not label Elav-marked hypocerebral neurons (*Figure 2—figure supplement 3D*). Additional studies are needed to distinguish the contributions of hypocerebral and pars intercerebralis Piezo neurons, with data so far suggesting that both subtypes of Piezo neurons contribute to feeding control.

## Piezo enteric neurons respond to crop-distending stimuli

Next, we investigated the response properties of Piezo-expressing enteric neurons. We analyzed neuronal activity using a transcriptional reporter system involving CaLexA through which sustained neural activity drives expression of GFP (*Masuyama et al., 2012*). CaLexA reporter was expressed in Piezo neurons using Gal4 drivers, along with an orthogonal activity-independent CD8-RFP reporter for normalization. For validation and determination of response kinetics, Trpa1-induced activation of Piezo neurons increased CaLexA reporter levels gradually, with maximal induction by 24 hr (*Figure 3—figure supplement 1A*). First, we asked whether hypocerebral Piezo neurons, and for comparison hypocerebral Gr43a neurons that function as peripheral sugar sensors, changed activity with feeding state (*Figure 3A, B*). For both neuron types, we observed that CaLexA-driven GFP

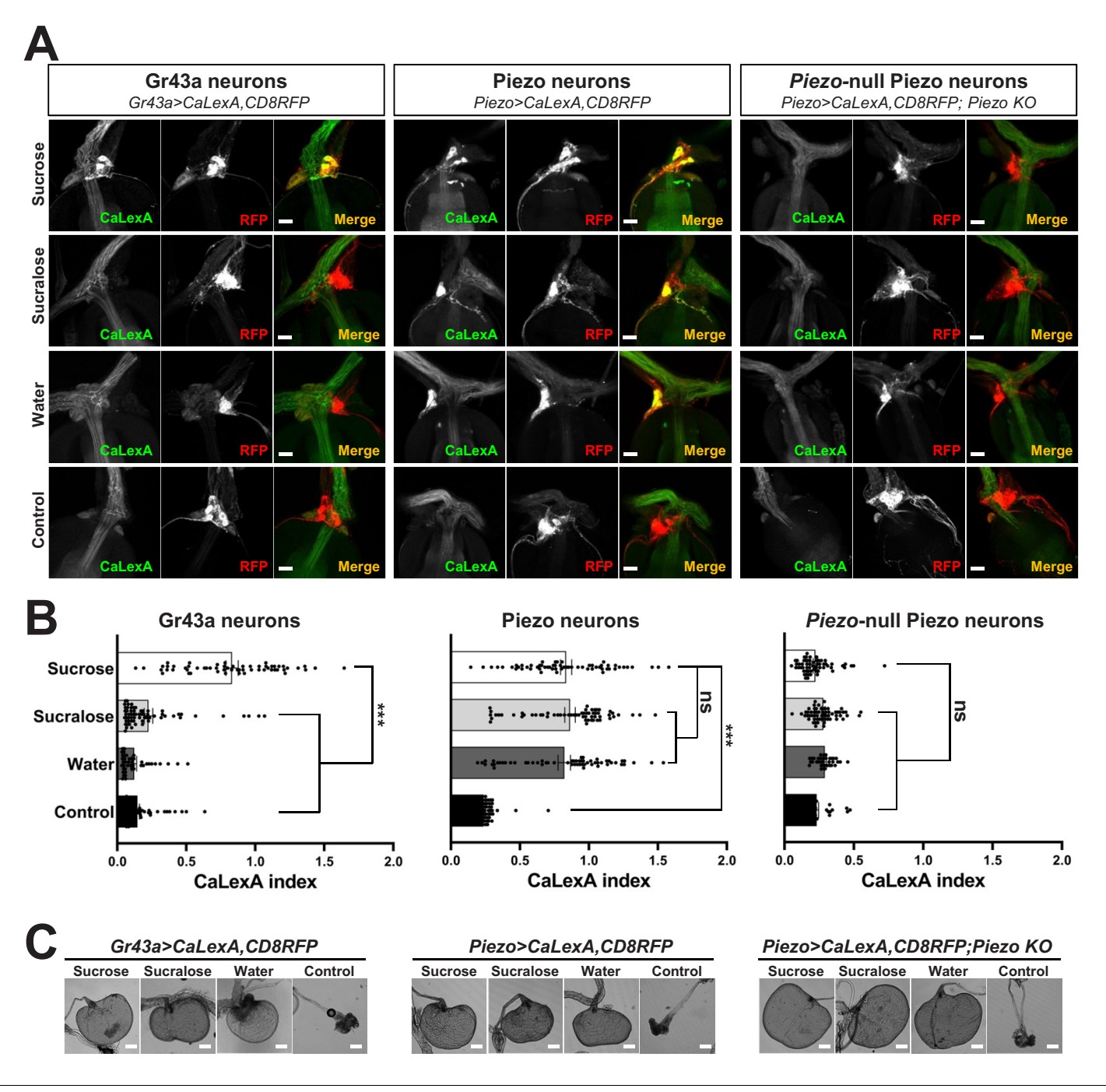

**Figure 3.** Piezo mediates enteric neuron responses to crop-distending stimuli. (**A**) Flies of genotypes indicated were provided solutions of (1) sucrose, (2) sucralose, (3) water alone after a period of water deprivation (water), or (4) water alone ad libitum for 24 hr (control). Representative images of native CaLexA-induced GFP reporter (green) and CD8RFP (red) fluorescence visualized in enteric Gr43a neurons (left), Piezo neurons (middle), or Piezo neurons lacking *Piezo* (right), scale bar 10 μm. (**B**) Quantification of CaLexA-induced GFP fluorescence in individual RFP-expressing neurons from flies in (**A**). n (from top to bottom): 59, 64, 43, and 67 Gr43a neurons from 13, 14, 9, and 15 flies; 61, 61, 59, and 66 Piezo neurons from 11, 11, 10, and 12 flies; 60, 60, 33, and 37 *Piezo*-null Piezo neurons from 11, 11, 5, and 6 flies, mean ± SEM, ***p<0.0001, ns: not significant by ANOVA Dunnett's multiple comparison test. (**C**) Visualization of the crop from flies given stimuli indicated after 24 hr (sucrose, sucralose, control) or 15 min (water), scale bar 100 μm. See *Figure 3—figure supplement 1* and source data.

The online version of this article includes the following source data and figure supplement(s) for figure 3:

**Source data 1.** Numerical data to support the graph in *Figure 3*.

*Figure 3 continued on next page*

Figure 3 continued

**Figure supplement 1.** Responses and innervation patterns of Piezo neurons in wild-type and Piezo knockout flies.

**Figure supplement 1—source data 1.** Numerical data to support the graph in *Figure 3—figure supplement 1*.

expression was low after a fast or in flies fed ad libitum, but was strikingly elevated when flies engorged themselves on a sucrose diet (*Figure 3A, B*, *Figure 3—figure supplement 1B*). Sucrose consumption could potentially stimulate both gut chemosensors and mechanosensors as an increase in crop volume was observed compared with flies fed ad libitum (*Figure 3—figure supplement 1C*). We next asked whether activity changes in enteric neurons depended on the content of ingested material. We compared CaLexA-mediated GFP expression levels in flies fed for 24 hr with (1) sucrose, (2) sucralose, a sweetener that lacks caloric value and stimulates peripheral gustatory receptors but not internal Gr43a neurons, (3) water alone after a period of water deprivation, or (4) water alone ad libitum. Flies extensively consumed sucrose, sucralose, and water when water-deprived, resulting in acute increases in crop volume that were not observed in flies given only water ad libitum (*Figure 3C*). Enteric Gr43a neurons displayed elevated levels of CaLexA-mediated GFP expression after engorgement on sucrose, which is converted into fructose and glucose, but not sucralose or water, consistent with a role for these neurons in sensing nutritional carbohydrates (*Miyamoto and Amrein, 2014*). In contrast, enteric Piezo neurons were activated more generally by sucrose, sucralose, and deprivation-induced water ingestion, but not in controls given only water ad libitum, with responses correlated to the extent of gut distension. The observation that Piezo neurons were similarly activated by water- and sucrose-induced gut distension indicated a sensory mechanism that does not require chemosensation of particular nutrients. Together, these findings suggest a model of two segregated sensory pathways through the hypocerebral ganglion, with Gr43a neurons responding to sugars and Piezo neurons responding to anterior gut mechanosensation.

## *Piezo* knockout alters enteric neuron responses and fly feeding behavior

Next, we asked whether the Piezo receptor mediates neuronal responses of hypocerebral neurons. We obtained Piezo knockout flies and crossed them with flies harboring alleles, enabling the CaLexA reporter system in Piezo neurons (using *Piezo-Gal4$^{59266}$* flies with the *Piezo-Gal4* transgene remote from the endogenous Piezo locus). Remarkably, hypocerebral ganglion neurons marked in *Piezo-Gal4* flies but lacking *Piezo* expression did not respond to engorgement by sucrose, sucralose, or water, even though the crops of *Piezo* knockout flies were distended (*Figure 3A, B*). (As shown below, the extent of distension is actually more pronounced in *Piezo* knockout flies, yet CaLexA-mediated responses were not observed.) A lack of neuronal responses in *Piezo* knockout flies is not due to gross deficits in the ability to produce reporter as Trpa1-mediated activation of Piezo neurons in Piezo knockout flies was sufficient to induce a CaLexA-mediated response (*Figure 3—figure supplement 1D*). Furthermore, Piezo neurons still innervated the anterior gut, suggesting that the deficit was not due to coarse developmental miswiring (*Figure 3—figure supplement 1E*). Instead, enteric neurons of *Piezo* knockout flies seemingly fail to respond to crop-distending stimuli due to a mechanosensory defect.

Next, we asked whether *Piezo* knockout flies display changes in behavior or physiology. We measured feeding behavior in *Piezo* knockout flies and, for comparison, isogenic $w^{1118}$ flies. For synchronization, flies were fasted for 18 hr and then given ad libitum access to dye-labeled food for 30 min. Remarkably, *Piezo* knockout flies increased food intake and had visually observable crop distension (*Figure 4A–C*, *Figure 4—figure supplement 2A*). Moreover, *Piezo* knockout flies fed ad libitum on normal fly food for 5–7 days showed an increase in body weight compared to control flies (*Figure 4D*). Automated analysis of feeding patterns was performed involving an EXPRESSO platform (*Yapici et al., 2016*), and *Piezo* knockout flies displayed an increase in food intake and feeding bout duration but a similar frequency of feeding bout initiation (*Figure 4E*). Abnormal gut distension and feeding behavior were rescued by exogenous expression of Piezo-GFP in *Piezo* knockout neurons driven by *Piezo-Gal4* (*Figure 4F*, *Figure 4—figure supplement 1A*). Unlike Drop-dead knockout flies that have an enlarged crop due to defective food passage into the intestine (*Peller et al.,*

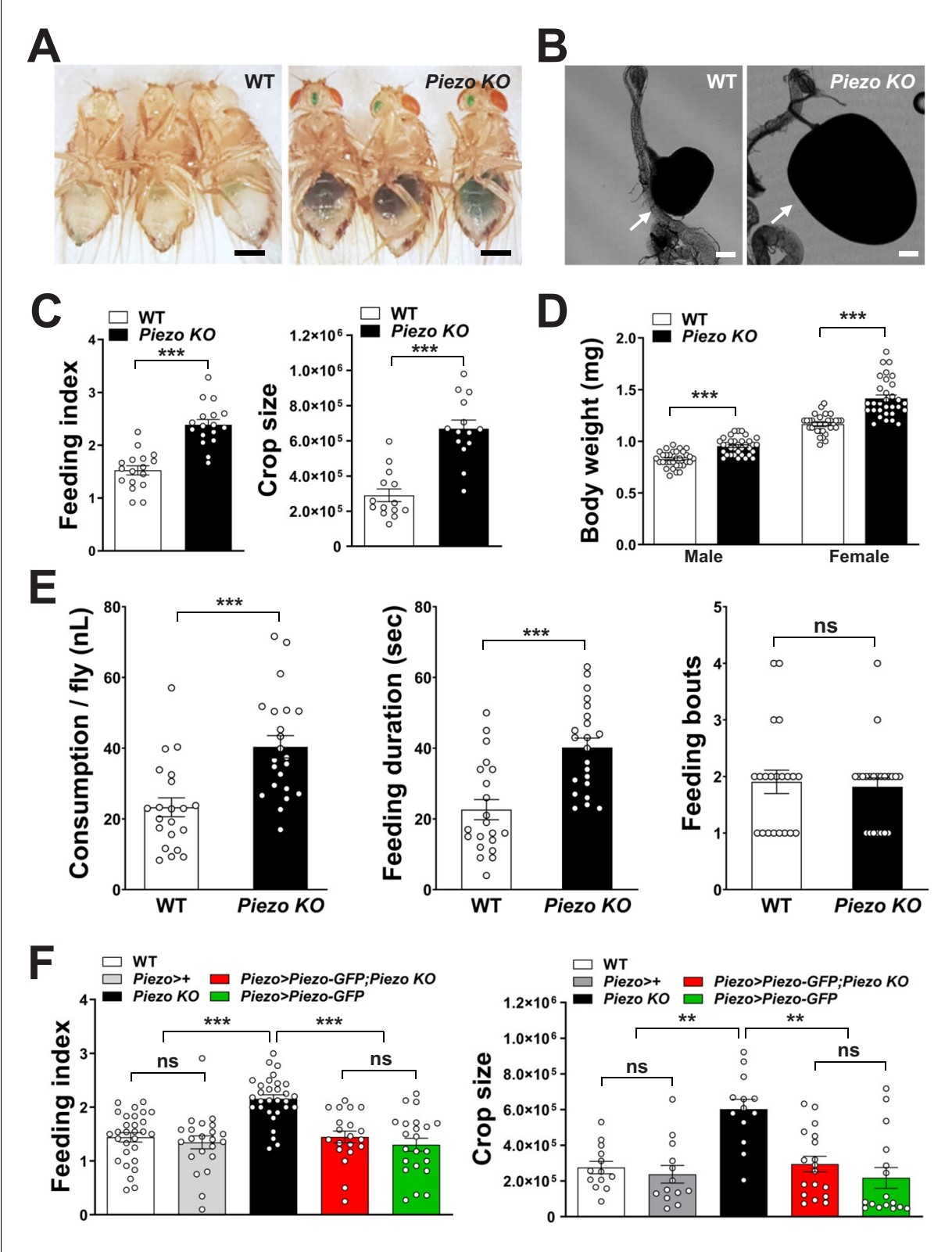

**Figure 4.** *Piezo* knockout alters fly feeding behavior. (**A**) Fasted wild-type (WT) and *Piezo* knockout (KO) female flies were given brief access (30 min) to dye-colored food and imaged, scale bar 0.5 mm. (**B**) Representative images of the crop (arrow) in WT and *Piezo* KO flies, scale bar 100 μm, (**C**) Calculated feeding indices (left) and crop sizes (right) from flies in (**A**). n (feeding index: 17 trials involving 204 flies), n (crop size): 14 flies, mean ± SEM, ***p<0.0001 by unpaired t-test. (**D**) Body weights of WT and *Piezo* KO flies fed regular food ad libitum. n (left to right): 32, 34, 33, and 31 trials involving

*Figure 4 continued on next page*

*Figure 4 continued*

three flies per trial, mean ± SEM, ***p<0.0001 by unpaired t-test. (**E**) Feeding parameters of fasted WT and *Piezo* KO male flies were analyzed using the EXPRESSO assay for 30 min after food introduction to determine overall food consumption, feeding duration per bout, and the number of bouts. n: 21 (WT), 22 (PIEZO KO) flies, mean ± SEM, ***p<0.0005, ns: not significant by unpaired t-test. (**F**) Calculated feeding indices (left) and crop sizes (right) from Piezo rescue and control flies indicated. n (left to right) (feeding index): 29, 22, 30, 22, and 13 trials involving 12 flies per trial. n (crop size): 13, 13, 13, 18, and 16 flies, mean ± SEM, ***p<0.0005, **p<0.005 by ANOVA Dunnett's multiple comparison test, ns: not significant by unpaired t test. See *Figure 4—figure supplements 1* and *2* and source data.

The online version of this article includes the following source data and figure supplement(s) for figure 4:

**Source data 1.** Numerical data to support the graph in *Figure 4*.
**Figure supplement 1.** Physiological characterization of Piezo knockout flies.
**Figure supplement 1—source data 1.** Numerical data to support the graph in *Figure 4—figure supplement 1*.
**Figure supplement 2.** Feeding characteristics of Piezo knockout flies.
**Figure supplement 2—source data 1.** Numerical data to support the graph in *Figure 4—figure supplement 2*.

*2009*), *Piezo* knockout flies have normal food transit, a normal lifespan, and increased defecation rates, presumably due to increased feeding (*Figure 4—figure supplement 1B–E*). Other than food-induced distension, the anatomy of the crop appeared normal in *Piezo* knockout flies as visualized by histology of crop muscle, analysis of cell density, and volume measurements during starvation (*Figure 4—figure supplement 1F–H*). As mentioned above, knockout of *Piezo* does not impact the extent of gut innervation (*Figure 3—figure supplement 1E*); furthermore, thermogenetic Trpa1-mediated activation of Piezo neurons in *Piezo* knockout flies suppressed feeding behavior (*Figure 4—figure supplement 1I*), indicating that neural circuits downstream of enteric Piezo neurons were intact and remained capable of eliciting a behavioral response after *Piezo* knockout. Piezo also functions to guide the differentiation of gut enteroendocrine cells from mechanosensitive ISCs (*He et al., 2018*); however, selectively restoring *Piezo* expression in ISCs using *Escargot-Gal4* (*Esg-Gal4*) did not rescue crop volume and feeding phenotypes (*Figure 4—figure supplement 1J*). We also note that while the crops of Piezo knockout flies are distended, the flies eventually stop eating (although abdomen bursting does rarely occur, *Figure 4—figure supplement 2B*), suggesting either a physical limitation to ingestion or eventual engagement of a secondary satiety pathway, perhaps through nutrient sensors or posterior gut mechanoreceptors. Taken together, our data indicate a role for Piezo in sensing anterior gut distension, and that disrupting the function of Piezo neurons, or Piezo itself, causes substantial changes to gut physiology and feeding behavior.

Food-induced gut distension is thought to be an evolutionarily conserved signal for meal termination, yet underlying mechanisms and sensory receptors have long remained mysterious. Furthermore, whether food volume sensors are required for normal feeding control has remained unknown as tools for selective loss of function were not available without knowing the underlying sensory mechanisms. Here, we reveal a role for *Drosophila* Piezo in neurons that innervate the anterior gut and sense the size of a meal. Disrupting this pathway increases food consumption and body weight, and causes swelling of the gastrointestinal tract. These studies demonstrate that anterior gut mechanosensation contributes to the complex calculus that underlies the decision to eat, and provide a foundation for the comparative physiology and evolution of feeding control. Moreover, understanding related pathways in humans may enable new therapies for treating obesity and other food consumption disorders.

## Materials and methods

### Key resources table

| Reagent type (species) or resource | Designation | Source or reference | Identifiers | Additional information |
|---|---|---|---|---|
| Genetic reagent (*Drosophila melanogaster*) | *Piezo-Gal4* | Bloomington *Drosophila* Stock Center | BDSC: 59266; RRID:BDSC_59266 | |

*Continued on next page*

*Continued*

| Reagent type (species) or resource | Designation | Source or reference | Identifiers | Additional information |
|---|---|---|---|---|
| Genetic reagent (*D. melanogaster*) | *Piezo(KI)-Gal4* | *He et al., 2018* | PMID:29414942 | |
| Genetic reagent (*D. melanogaster*) | *Piezo(gene-trap)-Gal4* | Bloomington *Drosophila* Stock Center | BDSC: 76658 RRID:BDSC_76658 | |
| Genetic reagent (*D. melanogaster*) | *Piezo KO* | Bloomington *Drosophila* Stock Center | BDSC: 58770; RRID:BDSC_58770 | Isogenized with w[1118] |
| Genetic reagent (*D. melanogaster*) | *UAS-GFP-Piezo* | Bloomington *Drosophila* Stock Center | BDSC: 58773; RRID:BDSC_58773 | |
| Genetic reagent (*D. melanogaster*) | *UAS-CD8RFP* | Bloomington *Drosophila* Stock Center | BDSC: 32218; RRID:BDSC_32218 | |
| Genetic reagent (*D. melanogaster*) | *Hs-Flp, UAS-MCFO* | Bloomington *Drosophila* Stock Center | BDSC: 64085; RRID:BDSC_64085 | |
| Genetic reagent (*D. melanogaster*) | *UAS-CD8GFP* | Bloomington *Drosophila* Stock Center | BDSC: 5137; RRID:BDSC_5137 | |
| Genetic reagent (*D. melanogaster*) | *UAS-Trpa1* | Bloomington *Drosophila* Stock Center | BDSC: 26263; RRID:BDSC_26263 | |
| Genetic reagent (*D. melanogaster*) | *UAS-CaLexA* | Bloomington *Drosophila* Stock Center | BDSC: 66542; RRID:BDSC_66542 | |
| Genetic reagent (*D. melanogaster*) | *Nanchung-Gal4* | Bloomington *Drosophila* Stock Center | BDSC: 24903; RRID:BDSC_24903 | |
| Genetic reagent (*D. melanogaster*) | *Inactive-Gal4* | Bloomington *Drosophila* Stock Center | BDSC: 36360; RRID:BDSC_36360 | |
| Genetic reagent (*D. melanogaster*) | *Painless-Gal4* | Bloomington *Drosophila* Stock Center | BDSC: 27894; RRID:BDSC_27894 | |
| Genetic reagent (*D. melanogaster*) | *Tmc-Gal4* | *Zhang et al., 2016* | PMID:27478019 | |
| Genetic reagent (*D. melanogaster*) | *Gr43a-Gal4* | *Miyamoto et al., 2012* | PMID:23178127 | |
| Genetic reagent (*D. melanogaster*) | *Gr43a-LexA* | *Fujii et al., 2015* | PMID:25702577 | |
| Genetic reagent (*D. melanogaster*) | *UAS-DenMark* | Bloomington *Drosophila* Stock Center | BDSC: 33061; RRID:BDSC_33061 | |
| Genetic reagent (*D. melanogaster*) | *UAS-DenMark* | Bloomington *Drosophila* Stock Center | BDSC: 33062; RRID:BDSC_33062 | |
| Genetic reagent (*D. melanogaster*) | *Trp-Gal4* | Bloomington *Drosophila* Stock Center | BDSC: 36359; RRID:BDSC_36359 | |
| Genetic reagent (*D. melanogaster*) | *Nompc-Gal4* | Bloomington *Drosophila* Stock Center | BDSC: 36360; RRID:BDSC_36360 | |
| Genetic reagent (*D. melanogaster*) | *Drop-dead KO* | Bloomington *Drosophila* Stock Center | BDSC: 36360; RRID:BDSC_36360 | |

*Continued on next page*

*Continued*

| Reagent type (species) or resource | Designation | Source or reference | Identifiers | Additional information |
|---|---|---|---|---|
| Genetic reagent (*D. melanogaster*) | *w[1118]* | Bloomington *Drosophila* Stock Center | BDSC: 3605; RRID:BDSC_3605 | |
| Genetic reagent (*D. melanogaster*) | *Trpa1-Gal4* | Bloomington *Drosophila* Stock Center | BDSC: 36362; RRID:BDSC_36362 | |
| Genetic reagent (*D. melanogaster*) | *Ppk-Gal4* | Bloomington *Drosophila* Stock Center | BDSC: 32078; RRID:BDSC_32078 | |
| Genetic reagent (*D. melanogaster*) | *GMR51F12-Gal4* | Bloomington *Drosophila* Stock Center | BDSC: 58685; RRID:BDSC_58685 | |
| Genetic reagent (*D. melanogaster*) | *Cha-Gal80* | *Sakai et al., 2009* | PMID:19531155 | |
| Genetic reagent (*D. melanogaster*) | *UAS-Shibire[ts]* | *Kitamoto, 2001* | PMID:11291099 | |
| Genetic reagent (*D. melanogaster*) | *Escargot-Gal4* | *Hayashi et al., 2002* | PMID:12324948 | |
| Genetic reagent (*D. melanogaster*) | *Elav-Gal80* | *Yang et al., 2009* | PMID:19249273 | |
| Antibody | Anti-Dilp2; rabbit polyclonal | Veenstra Jan (University of Bordeaux, France) | | (1:200) |
| Antibody | Anti-GFP; chicken polyclonal | Thermo Fisher Scientific | Thermo Fisher Scientific Cat# A10262; RRID:AB_2534023 | (1:200) |
| Antibody | Anti-RFP; rabbit polyclonal | Rockland | Rockland Cat# 600-401-379; RRID:AB_2209751 | (1:200) |
| Antibody | Anti-Elav; mouse monoclonal | Developmental Studies Hydridoma Bank | DSHB Cat# Elav-9F8A9; RRID:AB_528217 | (1:200) |
| Antibody | Anti-Akh; rabbit polyclonal | Kerafast | Kerafast Cat# EGA261 | (1:200) |
| Antibody | Anti-Flag; Rat monoclonal | Novus Biologicals | Novus Cat# NBP1-06712SS; RRID:AB_1625982 | (1:200) |
| Antibody | Anti-HA; Rabbit monoclonal | Cell Signaling Technology | Cell Signaling Technology Cat# 3724S; RRID:AB_1549585 | (1:200) |
| Antibody | Anti-V5; Mouse monoclonal | Bio-Rad | Bio-Rad Cat# MCA2894D549GA RRID:AB_10845946 | (1:200) |
| Antibody | Alexa Fluor-488; Chicken polyclonal | Jackson ImmunoResearch | Jackson ImmunoResearch Cat# 703-545-155; RRID:AB_2340375 | (1:400) |
| Antibody | Alexa Fluor-488; Rabbit polyclonal | Jackson ImmunoResearch | Jackson ImmunoResearch Cat# 711-545-152; RRID:AB_2313584 | (1:400) |

*Continued*

| Reagent type (species) or resource | Designation | Source or reference | Identifiers | Additional information |
|---|---|---|---|---|
| Antibody | Cy3-AffiniPure; Rabbit polyclonal | Jackson ImmunoResearch | Jackson ImmunoResearch Cat# 711-165-152; RRID:AB_2307443 | (1:400) |
| Antibody | Alexa Fluor 647; Rabbit polyclonal | Jackson ImmunoResearch | Jackson ImmunoResearch Cat# 711-605-152; RRID:AB_2492288 | (1:400) |
| Antibody | Alexa Fluor 647; Mouse polyclonal | Jackson ImmunoResearch | Jackson ImmunoResearch Cat# 715-605-150; RRID:AB_2340862 | (1:400) |
| Antibody | Alexa Fluor 488; Mouse polyclonal | Jackson ImmunoResearch | Jackson ImmunoResearch Cat# 715-545-150; RRID:AB_2340846 | (1:400) |
| Antibody | Alexa Fluor 488; Rat polyclonal | Jackson ImmunoResearch | Jackson ImmunoResearch Cat# 712-545-153; RRID:AB_2340684 | (1:400) |
| Chemical compound, drug | Normal goat serum | Jackson ImmunoResearch | Jackson ImmunoResearch Cat# 005-000-121; RRID:AB_2336990 | (5%) |
| Chemical compound, drug | Fluoromount-G | Southern Biotech | 0100-01 | |
| Chemical compound, drug | Phalloidin-FITC | Sigma | P5282-1MG | (1:400) |
| Chemical compound, drug | Phalloidin-TRITC | Sigma | P1951-1MG | (1:400) |
| Chemical compound, drug | TO-PRO-3 | ThermoFisher | T3605 | (1:400) |
| Chemical compound, drug | Green food dye | Amazon | Amazon standard identification *number (ASIN):* B0055AFE5G | Manufacturer: McCormick |
| Software, algorithm | Prism 8 | GraphPad | RRID:SCR_002798 | |
| Software, algorithm | Fiji | Schindelin et al., Nature Methods, 2012 | PMID:22743772 | https://imagej.net/Fiji |
| Software, algorithm | Python-based custom data analysis code used for EXPRESSO assay | Samuel C. Whitehead, 2021, PiezoPaper ExpressoCode | | https://github.com/scw97/PiezoPaperExpressoCode; *Min, 2021*; copy archived at swh:1:rev:bd8a58fa0e4f796e2ed0b72fe807862305b84b6b |
| Other | Confocal microscope | Leica | Leica SP5 | |

## Flies

Fly stocks were maintained on a regular cornmeal agar diet (Harvard Exelixis facility) at 25°C, with mating and collection performed under $CO_2$ anesthesia. For *Piezo* knockout studies, *Piezo* knockout

flies were isogenized by outcrossing five times into a wild-type $w^{1118}$ isogenic background. We obtained *Piezo* knockout, knock-in (KI) *Piezo-Gal4* and *UAS-Piezo-GFP* flies (Norbert Perrimon), *Tmc-Gal4* (Craig Montell), knock-in *Gr43a-LexA* and knock-in *Gr43a-Gal4* (Hubert Amrein), and from Bloomington *Drosophila* Stock Center *Piezo-Gal4* (BDSC# 59266), Recombinase-Mediated Cassette Exchange (RMCE) gene-trap *Piezo-Gal4* (BDSC# 76658), *UAS-CD8GFP* (BDSC# 5137), *UAS-CD8RFP* (BDSC# 32218), *UAS-Trpa1* (BDSC# 26263), *Cha-Gal80* (BDSC# 60321), *UAS-CaLexA* (BDSC# 66542), *Nanchung-Gal4* (BDSC# 24903), *Inactive-Gal4* (BDSC# 36360), *Painless-Gal4* (BDSC# 27894), *Trp-Gal4* (BDSC# 36359), *Trpa1-Gal4* (BDSC# 36362), *Nompc-Gal4* (BDSC# 36361), *Ppk-Gal4* (BDSC# 32078), *UAS-DenMark* (BDSC# 33061 and 33062), *Drop-dead KO* (BDSC# 24901), $w^{1118}$ (BDSC# 3605), *GMR51F12-Gal4* (BDSC# 58685), and *Hs-Flp; UAS-MCFO* (BDSC# 64085). *Escargot-Gal4*, *Cha-Gal80*, *Elav-Gal80*, *UAS-Shibire^{ts}*, and *Dilp2-Gal4* were as published (*Hayashi et al., 2002*; *Ikeya et al., 2002*; *Kitamoto, 2001*; *Sakai et al., 2009*; *Yang et al., 2009*).

## Feeding analysis

Acute feeding assays were performed as previously described with modifications (*Albin et al., 2015*; *Min et al., 2016*). Twelve adult female flies were collected upon eclosion and housed in a vial with for 5–7 days. Prior to testing, baseline hunger was synchronized by starving flies for 15–18 hr in a vial containing only on a dampened kimwipe section. The surface of regular fly food (typically ~16.25 ml per 50 ml vial) was dyed with green food coloring (McCormick, 70 µl dye per vial) and dried (24 hr). For testing, starved flies were transferred to vials containing dyed food for 30 min. Trials were ended by cooling the vials on ice, and a feeding index was scored as described below (see quantification). For thermogenetic experiments, flies expressing Trpa1 or Shibire were maintained and starved at 18°C prior to testing. Ten minutes prior to testing, starved flies and dye-labeled food were pre-warmed to 30°C or 32°C for experiments with either Trpa1 or Shibire, and then tested as above. Feeding behavior was scored by visual inspection of ingested dye with scores given from 0 to 5 based on dye intensity, as reported previously (*Albin et al., 2015*; *Min et al., 2016*). A feeding index was expressed by averaging the feeding scores for all flies per vial (~12 flies). For automated analysis of feeding patterns, fasted male flies (3–5 days old) were individually introduced into chambers connected to an EXPRESSO machine (http://public.iorodeo.com/docs/expresso/hardware_design_files.html) and feeding bouts were analyzed using EXPRESSO acquisition software (http://public.iorodeo.com/docs/expresso/device_software.html). Briefly, flies were given access (30 min) to a 200 mM sucrose solution through a capillary, and capillary fluid volume was measured over time using the EXPRESSO instrument. Total food consumption, feeding duration, feeding bout numbers, and feeding latency were then calculated using a Python-based custom data analysis code available at https://github.com/scw97/PiezoPaperExpressoCode.

## Chronic studies of body weight, intestinal transit, fecal rate, and lifespan

Chronic studies were performed on 5–7-day-old male and female flies fed ad libitum with regular fly food. Flies were anesthetized (ice, 10 min) and weighed in groups of three in a 1.5-ml Eppendorf tube, with body weight expressed as the average weight per group of three. Lifespan was analyzed for a group of 12 flies by counting the number of surviving flies each day. Fecal rates were measured after feeding flies dye-colored food (dye-colored food is described above) for 1 hr, with visual inspection of abdominal dye to ensure ingestion. Flies were transferred to an empty vial containing a $1 \times 1$ cm filter paper floor for 30 min, and dye-labeled fecal spots on the filter paper were counted. For analysis of fecal deposition, individual data points reflect the mean behavior of ten flies. Intestinal transit was measured in flies given brief access (30 min) to dye-colored food, with dye location in the intestine determined visually. A transit index was calculated based on the leading dye edge position, with scores of 1, 2, and 3 referring to dye edge in the crop/anterior midgut, middle midgut, and hindgut/anus, respectively.

## Sparse neuronal labeling

*Piezo-Gal4* (59266) flies were crossed with MultiColor FlpOut (MCFO) flies (*Hs-Flp; UAS-MCFO* flies) that enable multicolor, stochastic, and sparse labeling of Gal4-expressing cells (*Nern et al., 2015*). MCFO flies contain multiple Gal4-dependent alleles encoding epitope tags, including HA, FLAG,

and V5. *Piezo-Gal4; Hs-Flp; UAS-MCFO* fly larvae were maintained at 19°C, and at the third instar, larvae (wandering stage) were heat-shocked (37°C, 15 min/day, 3 days) to induce reporter expression in dispersed neurons, and after eclosion, were collected for dissection of the brain and anterior gut and immunohistochemistry for HA, Flag, and V5 epitopes.

## Immunohistochemistry

Wholemount preparations of the gastrointestinal tract and brain were fixed (4% paraformaldehyde, phosphate buffered saline or PBS, 20 min, room temperature [RT]), washed (2 × 5 min, PBS with 0.5% Triton X-100), permeabilized (10 min, PBS with 0.5% Triton X-100), blocked (1 hr, RT, blocking solution: 5% normal goat serum [Jackson ImmunoResearch, 005-000-121], PBS with 0.1% Triton X-100), incubated with primary antibody (1:200, blocking solution, 4°C, overnight), washed (3 × 10 min, RT, PBS with 0.1% Triton X-100), incubated with secondary antibody (1:200, PBS with 0.1% Triton X-100, 2 hr, RT), washed (3 × 10 min, RT, PBS with 0.1% Triton X-100 then 2 × 5 min, RT, PBS), mounted on a slide glass with Fluoromount-G mounting medium (Southern Biotech, 0100-01), covered with a thin coverslip, sealed with nail polish, and analyzed by confocal microscopy (Leica SP5). Primary antibodies were anti-GFP (Thermo Fisher Scientific, Chicken, A10262), anti-RFP (Rockland, Rabbit, 600-401-379), anti-Elav (Developmental Studies Hydridoma Bank, Mouse, Elav-9F8A9), anti-Akh (Kerafast, Rabbit, EGA261), anti-Dilp2 (from Veenstra Jan, University of Bordeaux, France), anti-Flag (Novus Biologicals, Rat, NBP1-06712SS), anti-HA (Cell Signaling Technology, Rabbit, 3724S), and anti-V5 (Bio-Rad, Mouse, MCA2894D549GA). Secondary antibodies were anti-Chicken-Alexa Fluor-488 (Jackson ImmunoResearch, 703-545-155), anti-Rabbit-Alexa Fluor-488 (Jackson ImmunoResearch, 711-545-152), anti-Rabbit-Cy3 (Jackson ImmunoResearch, 711-165-152), anti-Rabbit-Alexa Fluor-647 (Jackson ImmunoResearch, 711-605-152), anti-Mouse-Alexa Fluor-647 (Jackson ImmunoResearch, 715-605-150), anti-Mouse-Alexa Flour-488 (Jackson ImmunoResearch, 715-545-150), and anti-Rat-Alexa Fluor 488 (Jackson ImmunoResearch, 712-545-153). For staining of visceral muscle and nuclei, Phalloidin-FITC (Sigma, P5282-1MG), Phalloidin-TRITC (Sigma, P1951-1MG), and TO-PRO-3 (ThermoFisher, T3605) were added together with the secondary antibody.

## Quantification of crop size and composition

After the feeding assay, flies were fixed (4% paraformaldehyde, PBS, RT, 1 hr) and decapitated. The anterior gastrointestinal tract was surgically removed after gentle displacement of appendages and thoracic muscles. Dissected tissue was washed (3× PBS, RT, 5 min) and mounted for bright-field microscopy using the 'Analyze-Measure' tool in Fiji to calculate crop area. Crop muscle and cell density were quantified as detailed below. For quantification of crop muscle density, the intensity of the Phalloidin-labeled muscle fibers in a region of interest (ROI) was divided by the total ROI area. For cell density, the number of nuclei labeled with TO-PRO-3 and counted using 'Analyse-3D objects counter' function in Fiji (https://imagej.net/Fiji) was divided by total ROI area.

## Analyzing neuronal responses with CaLexA

CaLexA responses were measured in *Piezo-Gal4* or *Gr43a-Gal4* flies containing *UAS-CaLexA* (*LexA-VP16-NFAT, LexAop-rCD2-GFP,* and *LexAop-CD8GFP-2A-CD8GFP*), and *UAS-CD8RFP*. Responses of Piezo knockout neurons were measured by introducing Piezo knockout alleles into *Piezo-Gal4; UAS-CaLexA; UAS-CD8RFP* flies. For sucrose and sucralose responses, flies were fed ad libitum with regular food, transferred to vials containing a kimwipe soaked with 10% sucrose solution or 1% sucralose solution containing green food coloring for 24 hr, and analyzed for crop distension and CaLexA expression. For water responses, flies were deprived of food and water for 6 hr, and transferred to vials containing a water-soaked kimwipe. Some flies were harvested after 15 min for analysis of crop distension and others were harvested after 18 hr for analysis of CaLexA expression. Control flies were placed in a vial containing a water-soaked kimwipe but no food for 24 hr and harvested for analysis. For TrpA1-mediated neuron stimulation, WT and *Piezo KO* flies bearing a *Piezo-Gal4*, *UAS-CaLexA* (*LexA-VP16-NFAT, LexAop-rCD2GFP,* and *LexAop-CD8GFP-2A-CD8GFP*), and *UAS-Trpa1* were placed in a 30°C incubator for 24 hr prior to analysis. For analysis of CaLexA expression, flies were anesthetized (ice, 10 min), and the anterior gastrointestinal tract was surgically removed. Dissected tissue was fixed (4% paraformaldehyde, PBS, 20 min, RT), washed (3 × 5 min, PBS), and slide mounted with Fluoromount-G mounting medium and a coverslip. Native GFP

(derived from CaLexA activation) and RFP (constitutive from a Gal4-dependent reporter) fluorescence was analyzed by confocal microscopy (Leica SP5).

For quantification of CaLexA-dependent reporter in *Figure 3B* and S3B, intensity of GFP and RFP fluorescence was calculated per neuron and a CaLexA index expressed as GFP fluorescence divided by RFP fluorescence. For quantification of CaLexA-dependent reporter in *Figure 2—figure supplement 2A, D*, which involved flies lacking an RFP allele for neuron identification and normalization, GFP intensity was measured in the whole hypocerebral ganglion and a background subtraction was performed involving a comparably sized region of the proventriculus lacking Gal4-positive cell bodies. For S3A and S3D, background-subtracted GFP fluorescence was divided by RFP fluorescence from a control *Piezo-Gal4; UAS-CD8RFP* fly to generate a CaLexA index.

## Statistical analysis

Data in graphs are represented as means ± SEM, with sample sizes provided in figure legends. Statistical significance was analyzed by ANOVA Dunnett's multiple comparison test or unpaired t-test using Prism 8 software (GraphPad), as indicated in figure legends.

## Acknowledgements

We thank Norbert Perrimon, Veenstra Jan, Bryan Song, and Dragana Rogulja for reagents and advice, Jinfei Ni for blinded analysis of behavior, Norbert Perrimon, Craig Montell, Julie Simpson, Hubert Amrein, and Bloomington *Drosophila* Stock Center for flies, and Hansine Heggeness and Exelixis facility at Harvard Medical School for fly food and stock maintenance.

## Additional information

### Competing interests

Stephen Liberles: Reviewing editor, *eLife*. The other authors declare that no competing interests exist.

### Funding

| Funder | Grant reference number | Author |
|---|---|---|
| American Heart Association | 20POST35210914 | Soohong Min |
| National Institutes of Health | NS090994 | David Van Vactor |
| National Institutes of Health | RO1DK116294 | Greg SB Suh |
| National Institutes of Health | RO1DK106636 | Greg SB Suh |
| Samsung Science and Technology Foundation | SSTF-BA-1802-11 | Greg SB Suh |
| Howard Hughes Medical Institute | | Stephen Liberles |
| National Institutes of Health | R35 GM133698-01 | Nilay Yapici |

The funders had no role in study design, data collection and interpretation, or the decision to submit the work for publication.

### Author contributions

Soohong Min, Conceptualization, Data curation, Formal analysis, Funding acquisition, Validation, Investigation, Visualization, Methodology, Writing - original draft, Writing - review and editing; Yang-kyun Oh, Data curation, Formal analysis, Investigation, Visualization, Methodology; Pushpa Verma, Resources; Samuel C Whitehead, Resources, Software, Funding acquisition; Nilay Yapici, Greg SB Suh, Conceptualization, Resources, Software, Supervision, Funding acquisition, Project administration, Writing - review and editing; David Van Vactor, Conceptualization, Resources, Supervision, Funding acquisition, Writing - original draft, Project administration, Writing - review and editing;

Stephen Liberles, Conceptualization, Resources, Software, Supervision, Funding acquisition, Writing - original draft, Project administration, Writing - review and editing

## Author ORCIDs
Soohong Min ![ORCID] https://orcid.org/0000-0003-0683-2935
Stephen Liberles ![ORCID] https://orcid.org/0000-0002-2177-9741

## Decision letter and Author response
Decision letter https://doi.org/10.7554/eLife.63049.sa1
Author response https://doi.org/10.7554/eLife.63049.sa2

## Additional files

### Supplementary files
• Transparent reporting form

### Data availability
All datapoints used are provided in figures and in a source data file.

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
