## [Decision Letter]

**Acceptance summary:**

The paper elegantly demonstrates that the mechanosensor Piezo is important in the gut for controlling meal size. It also describes the circuits that control this behavior. The requirement for Piezo in the hypocerebral ganglion to detect enlargement of the crop provides an overall mechanism for the regulation.

**Decision letter after peer review:**

Thank you for submitting your article "Control of feeding by Piezo-mediated gut mechanosensation in *Drosophila*" for consideration by *eLife*. Your article has been reviewed by three peer reviewers, and the evaluation has been overseen by a Reviewing Editor and Piali Sengupta as the Senior Editor. The reviewers have opted to remain anonymous.

The three reviewers have discussed the reviews with one another and the Reviewing Editor has drafted this decision to help you prepare a revised submission.

The three reviewers appreciated the significance of your work and the elegance of the manuscript with the discovery of Piezo in neurons that control internal feeding. The reviewers are therefore positive but would like you to clarify three points that you should be able to complete in the coming weeks.

The paper would benefit with a better identification of the neurons involved, especially, the localization of the cell bodies and the axons. This would help readers better appreciate the work, given that another paper has already described the same neurons.

Therefore, we would like you to:

1) Clarify which subset of Piezo-positive neurons are responsible for innervating the crop vs. the anterior midgut;

2) Determine to what extent *Cha-Gal80* inhibits *Pz-Gal4* expression in the *Dilp2-Gal4* neurons;

3) Clarify whether *Dilp2-Gal4* is expressed in the HCG neurons.

We realize that the first point might require work that would exceed a reasonable timeframe, especially given the relative urgency of the situation, but you might already have these experiments in hand. Points #2 and 3 should easily be done.

Reviewer #1:

Gut mechanosensation is critical for controlling meal size, but the molecules and circuits responsible have not been described in detail. Working in *Drosophila*, the authors show that the evolutionarily conserved mechanosensor Piezo is important for controlling meal size. Piezo mutants are found to over-consume when feeding, with the crop becoming distended, and Piezo is shown to be required for a subset of Piezo+ neurons in the hypocerebral ganglion (HCG) to become activated post-feeding, presumably responding to inflation of the crop. Overall, the study is well-performed and the conclusions are careful and largely justified by the data presented. The only potential weak spot is the assignment of Piezo's behavioral functions to the Piezo(+) neurons in the HCG, but the authors do note this. Overall, Identifying Piezo's role in feeding control is a significant advance.

The behavioral experiments and CaLexA studies are overall well-performed and support the conclusions drawn.

The identity of the Piezo-positive neurons responsible for innervating the crop vs. the anterior midgut should be clarified. The Piezo-positive neurons in the HCG respond to feeding, but it is difficult to tell whether these neurons innervate the anterior midgut, the crop or both structures. A clearer view of what this subset of Pz(+) neurons innervate would be useful in thinking about the mechanism at work. Perhaps a FLP-out stochastic labeling experiment or an intersectional labeling experiment could help clarify this issue.

Clarifying the extent to which *Cha-Gal80* blocks *Pz-Gal4* activity in the brain would be useful, as descending neurons could also contribute to the behavioral effects observed. In the main figure, it appears that *Cha-Gal80* eliminates *Pz-Gal4*-driven RFP expression throughout the brain, but in Figure 2—figure supplement 1 it appears that the IPC is spared. This is particularly interesting as the companion paper from Wang et al., 2020, identifies the *Dilp2-Gal4* neurons of the IPC rather than the HCG as Piezo's site of action in feeding control. It would be helpful to know: 1) to what extent *Cha-Gal80* inhibits *Pz-Gal4* expression in the *Dilp2-Gal4* neurons and 2) whether *Dilp2-Gal4* is expressed in the HCG neurons. These experiments should be straightforward to perform and would help provide a sense of the relative importance of HCG neurons vs. Dilp2 neurons in controlling feeding behavior.

Reviewer #2:

In the 70th, Dethier and Gelperin described mechanosensory nerves innervating the crop in large flies, which induced hyperphagia after transecting them. In this work, Min et al. demonstrate convincingly that similar neurons are present in *Drosophila*, which express a gene called Piezo. These stretch receptors innervate the crop and the anterior midgut of the digestive system of adult flies. The cells expressing Piezo are distinct from cells expressing genes involved in meal termination like gustatory receptors (Gr43a), the receptor to the adipokinetic hormone (Akh), and also distinct from other cells innervating the gut (like GMR41F12). Using genetic constructions targeting Piezo neurons that activate them (heat activated TRPA1 inducing action potentials) or inhibit them (temperature-sensitive Shibire blocking synaptic transmission at permissive temperatures), they show these neurons induce a cessation of the feeding activities. Piezo neurons thus modulate the volume of food ingested and more precisely, the volume stored into the crop. Flies with a deficient Piezo gene (KO), are hyperphagic, with an enormously distended crop, while flies in which this gene is rescued are comparable to control flies. Additional observations using a CaLexA reporter system confirm that feeding induce a functional activation of these neurons, and that these Piezo neurons are involved in sensing the state of distension of the crop. This work provides a solid foundation to a better understanding of the mechanosensory neurons that allow flies to control the volume of liquids ingested.

Reviewer #3:

This paper investigates the mechanism by which mechanosensory neurons that project to the gut sense that it is full of food. Using reporter lines labeling cells expressing various mechanotransduction channels, they identified expression of Piezo in hypocerebral ganglion neurons with mechanosensory endings in the gut. They show that silencing Piezo-expressing neurons promoted feeding while activating them inhibited feeding. Moreover, they showed using the CaLexA reporter that stimuli that distend the gut increased the activity of Piezo-expressing neurons, and that this activation was lost in Piezo knockout flies. The Piezo knockout mutants also showed behavioral abnormalities indicating that they no longer stop feeding when their gut is full. This evidence suggests Piezo is the mechanotransducer in the hypocerebral ganglion neurons which senses a distended gut.

Overall this is an interesting story that expands the range of functions for Piezo, and thus is of significant interest to the mechanotransduction field. There are places where ideally one would have wanted to see more precise genetic definition of the cellular site of action for Piezo (i.e. based on the authors' interpretations, in the hypocerebral ganglion neurons); I assume this is due to a lack of specific driver lines, time pressure from the competing manuscript or both (I am not a fly person so I don't know whether more specific drivers even exist). I think the authors are pretty clear about the evidence and don't overstate things, so I am happy for the paper to be published in this form.

---

## [Author Response]

The three reviewers appreciated the significance of your work and the elegance of the manuscript with the discovery of Piezo in neurons that control internal feeding. The reviewers are therefore positive but would like you to clarify three points that you should be able to complete in the coming weeks.The paper would benefit with a better identification of the neurons involved, especially, the localization of the cell bodies and the axons. This would help readers better appreciate the work, given that another paper has already described the same neurons.Therefore, we would like you to:1) Clarify which subset of Piezo-positive neurons are responsible for innervating the crop vs. the anterior midgut;

We added new data involving sparse labeling of Piezo-expressing cells to distinguish the innervation patterns of HCG neurons and Dilp2 neurons of the pars intercerebralis, which is the location of gut-innervating Piezo neurons reported in Wang et al., 2020. Briefly, we crossed *Piezo-Gal4* flies with MultiColor FlpOut (MCFO) flies (*Hs-Flp; UAS-MCFO* flies) that enable multicolor, stochastic, and sparse labeling of Gal4-expressing cells (Nern et al., 2015). We obtained several flies with expression in HCG neurons but not in the pars intercerebralis, and in these flies, we still observed fibers in the crop and anterior midgut, with individual neurons capable of displaying selective innervation of one of these targets. These new data are presented in Figure 1—figure supplement 1C and clearly indicate that Piezo neurons outside of the pars intercerebralis innervate the anterior gut. Further supporting this idea, we have other new data from *Piezo-Gal4; Cha-Gal80* flies (please see response to point #2) that provide similar results. All of these anatomical findings are consistent with our observations that HCG Piezo neurons respond to gut distension in a Piezo-dependent manner. These findings, when taken together with the recent Wang et al., 2020 paper, suggest a model of multiple gut-innervating Piezo neuron subtypes.

2) Determine to what extent Cha-Gal80 inhibits Pz-Gal4 expression in the Dilp2-Gal4 neurons;

We performed additional experiments to quantify Gal4-driven reporter expression in various neurons of *Piezo-Gal4; UAS-CD8RFP; Cha-Gal80* flies and *Piezo-Gal4; UAS-CD8RFP* flies. In control *Piezo-Gal4; UAS-CD8RFP* flies, we observed reporter expression per fly in 6.2 ± 0.5 HCG neurons and 4.9 ± 1.0 π neurons, 2.9 ± 0.7 of which express Dilp2. In *Piezo-Gal4; UAS-CD8RFP; Cha-Gal80* flies, we observed reporter expression per fly in 5.2 ± 0.5 HCG neurons and 1.1 ± 0.5 π neuron, 0.4 ± 0.3 of which express Dilp2 (about half of flies had one co-labeled neuron and half had zero). In flies that lack any reporter expression in pars intercerebralis Dilp2 neurons, we still observed labeled neurites in the anterior midgut and crop nerve, consistent with findings from stochastic labeling detailed above that neurons outside of the pars intercerebralis innervate these regions. These new findings are presented in Figure 2—figure supplement 3B and C. Furthermore, despite rare labeling of pars intercerebralis Dilp2 neurons in *Piezo-Gal4; Cha-Gal80* flies, activation of labeled neurons still inhibited feeding. These findings, as above, are consistent with a model of multiple gut-innervating Piezo neuron subtypes.

3) Clarify whether Dilp2-Gal4 is expressed in the HCG neurons.

*Dilp2-Gal4* does not drive reporter expression to HCG neurons (stained by Elav). We observed Dilp2-labeled fibers that pass through the HCG ganglion but not Dilp2-labeled HCG soma; these findings are presented in Figure 2—figure supplement 3D.

We realize that the first point might require work that would exceed a reasonable timeframe, especially given the relative urgency of the situation, but you might already have these experiments in hand. Points #2 and 3 should easily be done.Reviewer #1:Gut mechanosensation is critical for controlling meal size, but the molecules and circuits responsible have not been described in detail. Working in *Drosophila*, the authors show that the evolutionarily conserved mechanosensor Piezo is important for controlling meal size. Piezo mutants are found to over-consume when feeding, with the crop becoming distended, and Piezo is shown to be required for a subset of Piezo+ neurons in the hypocerebral ganglion (HCG) to become activated post-feeding, presumably responding to inflation of the crop. Overall, the study is well-performed and the conclusions are careful and largely justified by the data presented. The only potential weak spot is the assignment of Piezo's behavioral functions to the Piezo(+) neurons in the HCG, but the authors do note this. Overall, Identifying Piezo's role in feeding control is a significant advance.The behavioral experiments and CaLexA studies are overall well-performed and support the conclusions drawn.The identity of the Piezo-positive neurons responsible for innervating the crop vs. the anterior midgut should be clarified. The Piezo-positive neurons in the HCG respond to feeding, but it is difficult to tell whether these neurons innervate the anterior midgut, the crop or both structures. A clearer view of what this subset of Pz(+) neurons innervate would be useful in thinking about the mechanism at work. Perhaps a FLP-out stochastic labeling experiment or an intersectional labeling experiment could help clarify this issue.

Thank you for thoughts on the paper. We performed the stochastic labeling experiments requested, which have provided a nice additional dataset. As described above, we added new data involving sparse labeling of Piezo-expressing cells to distinguish the innervation patterns of HCG neurons and Dilp2 neurons of the pars intercerebralis, which is the location of gut-innervating Piezo neurons reported in Wang et al., 2020. Briefly, we crossed *Piezo-Gal4* flies with MultiColor FlpOut (MCFO) flies (*Hs-Flp; UAS-MCFO* flies) that enable multicolor, stochastic, and sparse labeling of Gal4-expressing cells (Nern et al., 2015). We obtained several flies with expression in HCG neurons but not in the pars intercerebralis, and in these flies, we still observed fibers in the crop and anterior midgut, with individual neurons capable of displaying selective innervation of one of these targets. These new data are presented in Figure 1—figure supplement 1C, and clearly indicate that Piezo neurons outside of the pars intercerebralis innervate the anterior gut. Further supporting this idea, we have other new data from *Piezo-Gal4; Cha-Gal80* flies (please see response to point #2) that provide similar results. All of these anatomical findings are consistent with our observations that HCG Piezo neurons respond to gut distension in a Piezo-dependent manner. These findings, when taken together with the recent Wang et al., 2020 paper, suggest a model of multiple gut-innervating Piezo neuron subtypes.

Clarifying the extent to which Cha-Gal80 blocks Pz-Gal4 activity in the brain would be useful, as descending neurons could also contribute to the behavioral effects observed. In the main figure, it appears that Cha-Gal80 eliminates Pz-Gal4-driven RFP expression throughout the brain, but in Figure 2—figure supplement 1 it appears that the IPC is spared. This is particularly interesting as the companion paper from Wang et al., 2020, identifies the Dilp2-Gal4 neurons of the IPC rather than the HCG as Piezo's site of action in feeding control. It would be helpful to know: 1) to what extent Cha-Gal80 inhibits Pz-Gal4 expression in the Dilp2-Gal4 neurons and 2) whether Dilp2-Gal4 is expressed in the HCG neurons. These experiments should be straightforward to perform and would help provide a sense of the relative importance of HCG neurons vs. Dilp2 neurons in controlling feeding behavior.

We have performed both experiments requested. As described above, we first performed additional experiments to quantify Gal4-driven reporter expression in central and peripheral neurons of *Piezo-Gal4; UAS-CD8RFP; Cha-Gal80* flies and *Piezo-Gal4; UAS-CD8RFP* flies. In control *Piezo-Gal4; UAS-CD8RFP* flies, we observed reporter expression per fly in 6.2 ± 0.5 HCG neurons and 4.9 ± 1.0 π neurons, 2.9 ± 0.7 of which express Dilp2. In *Piezo-Gal4; UAS-CD8RFP; Cha-Gal80* flies, we observed reporter expression per fly in 5.2 ± 0.5 HCG neurons and 1.1 ± 0.5 π neuron, 0.4 ± 0.3 of which express Dilp2 (about half of flies of one co-labeled neuron and half have zero). In flies that lack any reporter expression in pars intercerebralis Dilp2 neurons, we still observed labeled neurites in the anterior midgut and crop nerve, consistent with findings from stochastic labeling detailed above that neurons outside of the pars intercerebralis innervate these regions. These new findings are presented in Figure 2—figure supplement 3B and C. Furthermore, despite rare labeling of pars intercerebralis Dilp2 neurons in *Piezo-Gal4; Cha-Gal80* flies, activation of labeled neurons still inhibited feeding. We also performed experiments that show that *Dilp2-Gal4* does not drive reporter expression to HCG neurons (stained by Elav). We observed Dilp2 fibers that pass through the HCG ganglion but not Dilp2-labeled HCG soma; these findings are presented in Figure 2—figure supplement 3D. Together, these findings are further consistent with a model of multiple gut-innervating Piezo neuron subtypes.

Reviewer #3:This paper investigates the mechanism by which mechanosensory neurons that project to the gut sense that it is full of food. Using reporter lines labeling cells expressing various mechanotransduction channels, they identified expression of Piezo in hypocerebral ganglion neurons with mechanosensory endings in the gut. They show that silencing Piezo-expressing neurons promoted feeding while activating them inhibited feeding. Moreover, they showed using the CaLexA reporter that stimuli that distend the gut increased the activity of Piezo-expressing neurons, and that this activation was lost in Piezo knockout flies. The Piezo knockout mutants also showed behavioral abnormalities indicating that they no longer stop feeding when their gut is full. This evidence suggests Piezo is the mechanotransducer in the hypocerebral ganglion neurons which senses a distended gut.Overall this is an interesting story that expands the range of functions for Piezo, and thus is of significant interest to the mechanotransduction field. There are places where ideally one would have wanted to see more precise genetic definition of the cellular site of action for Piezo (i.e. based on the authors' interpretations, in the hypocerebral ganglion neurons); I assume this is due to a lack of specific driver lines, time pressure from the competing manuscript or both (I am not a fly person so I don't know whether more specific drivers even exist). I think the authors are pretty clear about the evidence and don't overstate things, so I am happy for the paper to be published in this form.

Thank you for taking the time to review the paper and for your supportive comments. We have provided additional experiments, as detailed above, that show the extent of specificity in our genetic experiments involving *Cha-Gal80*, and suggest the model that distinct Piezo neuron subtypes contribute to feeding control.